# Carbon Fiber and Nickel Coated Carbon Fiber–Silica Aerogel Nanocomposite as Low-Frequency Microwave Absorbing Materials

**DOI:** 10.3390/ma13020400

**Published:** 2020-01-15

**Authors:** Agnieszka Ślosarczyk, Łukasz Klapiszewski, Tomasz Buchwald, Piotr Krawczyk, Łukasz Kolanowski, Grzegorz Lota

**Affiliations:** 1Institute of Building Engineering, Faculty of Civil and Transport Engineering, Poznan University of Technology, Piotrowo 3, PL-60965 Poznan, Poland; 2Institute of Chemical Technology and Engineering, Faculty of Chemical Technology, Poznan University of Technology, Berdychowo 4, PL-60965 Poznan, Poland; lukasz.klapiszewski@put.poznan.pl; 3Institute of Materials Research and Quantum Engineering, Faculty of Materials Engineering and Technical Physics, Poznan University of Technology, Piotrowo 3, PL-60965 Poznan, Poland; tomasz.buchwald@put.poznan.pl; 4Institute of Chemistry and Technical Electrochemistry, Faculty of Chemical Technology, Poznan University of Technology, Berdychowo 4, PL-60965 Poznan, Poland; piotr.krawczyk@put.poznan.pl (P.K.); lukasz.kolanowski@doctorate.put.poznan.pl (Ł.K.); grzegorz.lota@put.poznan.pl (G.L.)

**Keywords:** carbon fiber, nano-structures, porosity, fiber/matrix bond, electromagnetic interference

## Abstract

Silica aerogel-based materials exhibit a great potential for application in many industrial applications due to their unique porous structure. In the framework of this study, carbon fiber and nickel coated carbon fiber–silica aerogel nanocomposites were proposed as effective electromagnetic shielding material. Herein, the initial oxidation of the surface of carbon fibers allowed the deposition of a durable Ni metallic nanolayer. The fibers prepared in this way were then introduced into a silica aerogel structure, which resulted in obtaining two nanocomposites that differed in terms of fiber volume content (10% and 15%). In addition, analogous systems containing fibers without a metallic nanolayer were studied. The conducted research indicated that carbon fibers with a Ni nanolayer present in the silica aerogel structure negatively affected the structural properties of the composite, but were characterized by two-times higher electrical conductivity of the composite. This was because the nickel nanolayer effectively blocked the binding of the fiber surface to the silica skeleton, which resulted in an increase of the density of the composite and a reduction in the specific surface area. The thermal stability of the material also deteriorated. Nevertheless, a very high electromagnetic radiation absorption capacity between 40 and 56 dB in the frequency range from 8 to 18 GHz was obtained.

## 1. Introduction

The intense development of computers and information technology systems, as well as the increasing number of electric and electronic devices and their miniaturization during recent years has resulted in a new ecological threat for the environment. This is electromagnetic radiation and interference of waves with a wide frequency spectrum [1,2], and this tendency is visible both in the range of radio frequencies (3 kHz–300 MHz) and in microwave frequencies (300 MHz–100 GHz). Electromagnetic radiation and interference of waves at different frequencies can negatively influence both devices, as well as people and other living organisms present in the direct vicinity of the electromagnetic field. Therefore, in the recent years, special attention has been directed to the search for materials that could reduce or eliminate the negative consequences of electromagnetic radiation.

A preferable method of high protection shielding is based on the reflection or absorption of electromagnetic radiation. Here, the effectiveness of shielding strongly depends on the properties of the material and parameters of the electromagnetic wave [3]. Materials that employ a reflection mechanism are characterized by high electrical conductivity and include charge carriers (electrons and/or holes), which can react with the electromagnetic wave by damping or weakening it. Most often, such materials include metals, carbon materials, or metallic and/or carbon layers on or in other materials, such as polymers. The main disadvantage of metal-based applications is associated with their high density and low resistance to corrosion (mostly in case of steel), or low resistance to impacts (in case of aluminum). Therefore, there is a necessity to develop other, lighter materials, and polymers are one of the potential solutions.

Due to the fact that polymers act as insulators, they must be covered with a conductive layer of metal or they must contain a filler with conductive properties (for example metal particles, microspheres covered with metallic or carbon layer, etc.). There is also a small group of conductive polymers, among which the polyaniline (PANI), polythiophene (PTh), or polypyrrole (PPy) are most often used, for shielding electromagnetic radiation. Unlike metals or carbon materials, conductive polymers damp the electromagnetic radiation both according to the mechanism of reflection and that of absorption [4,5].

Absorption is the second mechanism of damping the electromagnetic radiation. Materials that exhibit such properties include electric or magnetic dipoles in their structure that can react with the electromagnetic wave, absorbing and weakening it. These dipoles can include different types of metal alloys in the following compositions: Ni (79.5%)–Fe (14%)–Cu (5%)–Cr (1.5%), magnesium alloys, compounds characterized by high dielectric constant (such as barium titanate BaTiO_3_), or metal oxides (for example, iron (III) oxide) that are characterized by a dipole moment and magnetic properties [4].

Another damping mechanism is based on multiple reflection. This involves weakening of the electromagnetic radiation through multi-reflective interaction with planes and interphase areas localized within the material structure. Materials that meet these criteria possess a porous structure with a developed specific surface area. This includes lightweight materials, e.g., foams (polymer, carbon ones), and other porous composites that contain more than one filler.

Recently, nanofillers—such as particles of nanosilver or other metals, nanotubes or metallic nanowires [6,7,8,9], carbon nanotubes [10], or graphene and graphene oxide [11]—have been used as fillers in such materials. Nanofillers with a high aspect ratio create a unique net structure in the material with high electrical conductivity. This process occurs easier than in the case of microfillers and results in very good damping properties [4,12]. This group of materials also includes the nanomaterials proposed in this publication—silica aerogel–carbon fibers.

Silica aerogels are made of air in 95% of their structure, the rest is silica crystal structure (SiO_2_). They have an open porous structure composed of particles with diameter less than 10 nm and pores smaller than 50 nm. These properties give silica aerogels very low density of 0.03–0.35 g/cm^3^, with fairly high specific surface area (500–1200 m^2^/g). They also have a thermal conductivity coefficient that is below 0.02 W/(m∙K) (classified as ‘low’), low dielectric constant of 1.1, speed of sound of 100 m/s, and refractive index within the range 1.0–1.08 [13,14]. The outstanding properties of silica aerogels allow for their use in numerous industries; they can be employed as thermal and acoustic insulators [15,16], catalytic carriers [17,18], absorbents of gases, liquids or energy and sensors [19,20,21]. However, apart from all the positive properties described above, silica aerogels have also drawbacks, the most important of which is their brittleness. The mechanical properties of silica aerogels can be modified by incorporating various additives to their structure, e.g., nanoparticles and metal nano-oxides or by applying reinforcement in the form of a short structural fiber or fiber mats [22,23,24].

In the presented results, unmodified silica aerogel and incorporated nickel-coated carbon fibers were tested to discern their properties. To date, such a solution has not been studied in terms of application in materials used for shielding electromagnetic radiation.

The main aim of the research was to obtain new lightweight silica aerogel-based composites with carbon fiber and nickel-coated carbon fiber reinforcement that could be used as potential protective material against electromagnetic radiation. Thus, we tested the shielding effectiveness in terms of transmission and reflection characteristics of silica aerogel with unmodified and nickel-coated carbon fiber in low-frequency microwave ranges (from 8 to 18 GHz).

## 2. Materials and Methods

### 2.1. Electrochemical Plating of Nickel Cover on Carbon Fiber Surface

Electrochemical plating of nickel cover on carbon fibers was conducted in a three-electrode system, by way of a DC power supply method—the galvanostatic method. In this method, the working electrode has a constant density value of electric current intensity, which in this case was equal to 10 mA/g. Carbon fibers were used as a working electrode, while Hg/Hg_2_SO_4_/1M H_2_SO_4_ and porous nickel plate were used as a reference and counter electrode, respectively. Another very important parameter in this process was the deposition time, which was equal to 20 h. The composition of the electrolyte is another complementary information about the process—the so-called bath, in which the process of electrochemical deposition of nickel onto the carbon fibers took place. The bath composition was as follows: NiSO_4_·7H_2_O, NiCl_2_·6H_2_O and H_3_BO_3_ (Chempur, Piekary Śląskie, Poland).

### 2.2. Preparation of Carbon Fiber–Silica Aerogel Nanocomposite

Water glass (Chempur, Piekary Śląskie, Poland), which is cheaper than organosilicon compounds, was used as a silica aerogel precursor. Ambient pressure drying was applied, excluding the more expensive supercritical drying, which is also more difficult to employ on an industrial scale. Since there is possibility of using the obtained materials in the construction industry, it was decided that the synthesis of silica aerogel will be conducted in the presence of a cheap acidic catalyst, i.e., citric acid (Chempur, Piekary Śląskie, Poland). The literature study confirmed that the structural parameters of silica aerogels synthesized are better in the presence of weak organic acids than in case of strong inorganic acids, e.g., hydrochloric acid, the presence of which may negatively affect other construction materials, such as concrete or steel [25]. The used carbon fibers were obtained using coal tar pitch, a much cheaper carbon precursor, which is characterized by a relatively high tensile modulus to bulk density ratio, good heat stability up to 600 °C, compatibility, and good electrical conductivity. The fibers were 700 µm long with a diameter of 13 µm (DONACARBO, Osaka, Japan). Carbon fibers were modified using nitric acid to increase the content of functional groups containing oxygen (e.g., hydroxyl, carboxyl) on their surface and to enhance the number of connections between the silica structure and carbon fibers according to procedure described earlier [26,27,28]. In case of nickel-coated carbon fibers, the oxidized surface resulted in improved electrodeposition of metal. Silica aerogel and its composites with the pure and nickel-coated carbon fibers were obtained using a 10-percent water solution of sodium silicate in the presence of citric acid at the concentration of 1.0 M. Fibers were added to the sodium silicate solution before gelation in the amounts of 10 and 15 vol.% and mixed using the electromagnetic stirrer to obtaining the homogeneous suspension. After gelling, the composites were introduced into a water and methanol mixture at 1:1 volume ratio for 24 h, and then in methanol for one week. Then the modification of the gel structure was carried out in a mixture of trimethylchlorosilane TMCS/n-hexane at a 1:4 volume ratio for 24 h, at the temperature of 50 °C (Sigma-Aldrich, Munich, Germany/Chempur, Piekary Śląskie, Poland). After this time, the composites were removed from the solution and air-dried.

### 2.3. Characterization Methods

Structural properties of silica aerogel nanocomposites with pure and nickel-coated carbon fibers were tested using the ASAP 2020 analyzer (Micromeritrics Co., Norcross, GA, USA). BET surface area (Brunauer–Emmett–Teller) was determined based on adsorption isotherms in low-temperature nitrogen sorption at −196 °C. The average pore diameters were calculated on the basis of nitrogen adsorption isotherms by way of the BET method, based on the 4 V/Å formula, where V stands for total pore volume determined in a single point of adsorption isotherm with *p*/*p*_0_ = 0.99. The bulk density (*ρ*_b_) was measured by the mass and volume of a cylindrical silica aerogel-based nanocomposite. The porosity was calculated by means of equation, where in the silica skeletal density (*ρ*_s_) was about 2.2 g/cm^3^:(1)$$Porosity\%=\left(1-\frac{\rho_{b}}{\rho_{s}}\right)$$

Characterization of surface properties of silica aerogel nanocomposite with pure and nickel-coated carbon fibers was performed by applying Fourier transform infrared spectroscopy (FTIR) in the 4000–450 cm^−1^ range. The spectra were obtained using a Vertex 70 spectrometer (Bruker Optik GmbH, Ettlingen, Germany). The analyzed materials were tested in the form of a pellet containing a mixture of anhydrous KBr (approx. 0.1 g) and 1 mg of the test substance. In order to prepare the tablets, the mixture was pressed in a special steel ring at a pressure of approximately 10 MPa under vacuum. In addition, a X-ray diffractometer (Malvern PANanalytical, Malvern, UK) was used to analyze the crystal structures of carbon fibers and metal coatings with CuKa radiation and a Ni filter.

Thermal stability analyses were carried out through application of the Jupiter STA 449F3 analyzer (Netzsch GmbH, Selb, Germany), based on the thermogravimetric method (TGA). The measurement consisted of heating the appropriate sample mass in a temperature range of 30–1000 °C, with a step of 10 °C/min, under a nitrogen atmosphere.

Raman spectroscopy analysis was then conducted using an inVia Renishaw microscope (Renishaw, Gloucestershire, UK) equipped with laser emitting a 785 nm wavelength. Herein, 1200 L/mm diffraction grating was used. Raman spectra of study materials were collected in the backscattering geometry in the spectral range from 200 cm^−1^ to 3200 cm^−1^. The analyses of Raman spectra were performed by means of WiRE 3.4 (Renishaw) software.

Microstructural analysis of carbon fibers after electrochemical deposition and silica aerogel–carbon fibers nanocomposite were carried out by means of scanning electron microscopy using a Tescan–3–Vega microscope (Tescan, Brno, Czech Republic). In addition energy-dispersive X-ray spectroscopy (EDS) based on Brucker equipment was carried out. The conductivity of the obtained materials was studied in a Swagelok^®^ (Solon, OH, USA) cell by means of electrochemical impedance spectroscopy (EIS) and linear sweep voltammetry (LSV) techniques through utilization of the potentiostat/galvanostat VMP3 (Bio-Logic, Seyssinet-Pariset, France). EIS measurements were conducted at 0.2 V for 1 kHz with 10 mV amplitude, while LSV was conducted with a 2 mV/s scan rate from 0 to 0.2 V.

The reflection and transmission coefficients for electromagnetic shielding effectiveness of the tested nanocomposites were measured in the range of frequencies between 8 and 18 GHz by employing a system constructed in the Military Institute of Armament Technology in Zielonka (Poland). The signal generator inside the vector network analyzer created different frequencies of microwaves and carried these to the top sample holder by a coaxial cable. For testing, the nanocomposite sample with thickness of 5 mm was placed between the top and bottom sample holders. The received signal was processed through the corresponding port and finally, the transmission *T* and reflection *R* coefficients for electromagnetic radiation of the tested nanocomposite were assessed using the following equations:(2)$$T=10\log\left(\frac{P_{2}}{P_{1}}\right)\left[\mathrm{dB}\right]$$
(3)$$R=10\log\left(\frac{P_{3}}{P_{1}}\right)\left[\mathrm{dB}\right]$$
where:

*P*_1_ measured power [W] of electromagnetic wave directed at the sample,

*P*_2_ measured power [W] of electromagnetic wave after transition the sample,

*P*_3_ measured power [W] of electromagnetic wave reflected from the sample.

The absorption capability for electromagnetic radiation in % was calculated by the assumption that the sum *R + A + T* = 1.

## 3. Results and Discussion

### 3.1. Physical and Chemical Characterization of Carbon Fibers, Nickel-Coated Carbon Fibers, and Pure Silica Aerogel

Metal plating on carbon-based materials or other metal particles can be performed via chemical or electrochemical deposition [29,30,31]. In case of electrochemical plating the properties of the support are remarkably significant in terms of the effectiveness of metal electrodeposition.

It must be characterized by good electrical conductivity, wettability, and presence of active centers. In spite of this, the edges or surface defects are the best areas in the matrix at which the metal nucleation occurs and further layers of metal are developed. In order to increase the roughness of surface, the isotropic carbon fibers from coal tar pitch underwent modification in concentrated HNO_3_. After the oxidation treatment, the surface area of carbon fibers changed from 2 to 10 m^2^/g and exhibited hydrophilic character. In this process, the carbon fiber surface was enriched in its carboxylic and hydroxyl groups and promoted the further chemical reaction with silica gel hydroxide groups. This led to a homogeneous structure of silica aerogel nanocomposite. Moreover, initial oxidation of the carbon fiber surface resulted in an increased number of active centers and enabled the nucleation of nickel particles and deposition of a reasonable uniform layer of nickel (Figure 1a). As shown in Figure 1a, not all carbon fibers were entirely covered by nickel layer. This outcome could have come about by the electroplating process being carried out in a narrow pocket. Hence, some parts of the fibers had limited access to electrolyte. Still, the conductivity of carbon fibers after nickel electrodeposition was increased from 11.263 mS/cm to 13.269 mS/cm when tested by the LSV method (corresponding values as tested by the EIS method equaled 11.263 and 13.269 mS/cm, respectively). The presence of a nickel nanolayer on the carbon fiber surface was also confirmed by the EDS analysis presented in the Figure 1b,c, respectively, for pure and Ni-coated carbon fibers. The mapping of carbon and nickel elements for Ni-coated carbon fibers indicates a non-uniform layer of metal, however, it is also visible that in bundle of fibers there are particular fiber entirely covered by Ni-layer.

X-ray diffraction patterns for unmodified and Ni-coated carbon fibers are presented in Figure 2. Here, pure carbon fibers are characterized by low degree of carbonization. This was confirmed by diffuse peaks corresponding to graphite planes at (002) and (100) [32]. Calculated based on Bragg equation, that the interlayer distance equaled d_002_ = 0.353 nm confirmed the notion that the presented carbon fibers are isotropic in nature. For the nickel-coated carbon fibers, besides peaks for graphite, there are also well-developed peaks corresponding to planes (111), (200), and (220) in the nickel coating [33].

Silica aerogel was synthesized from 10 wt.% water solution of sodium silicate in the presence of 1.0 M citric acid as catalyst. Chemical modification in TMCS/n-hexane mixture enabled the ambient pressure drying of silica gel and creation a stable structure of silica aerogel in the form of granulate. The structural parameters of aerogel and morphology were presented in Table 1 and in Figure 3.

The silica aerogel is characterized by well-developed surface area 496.5 m^2^/g with micropores in the amount of 1.27 cm^3^/g and pores between silica nanoparticles of 10.2 nm. The density of pure silica aerogel equaled 0.201 g/cm^3^, while the porosity amounted to 91%. The morphology and microstructure of a prepared silica aerogel were analyzed by SEM and EDS analyses. Figure 3a–d show the view of the silica aerogel nanostructure and the mapping distribution of silicon and oxygen elements. Figure 3f presents the nitrogen adsorption-desorption isotherms at 77 K.

Based on the IUPAC classification and structural parameters obtained in BET analysis, it is obvious that the synthesized silica aerogel indicates a porous structure and belongs to the mesoporous-like material family. In addition, a confirmation of mesoporous structure in the shape of adsorption and desorption isotherms is shown in Figure 3f. According to the classification of porous materials made by de Boer in 1958 [34], from the shape of the desorption hysteresis loop it can be found that pore structures are both cylindrical capillary pores open at both ends and cylindrical pores closed at one end with a narrow neck at the other, like an “ink-bottle”, which are assigned to the materials with the mesoporous structure.

### 3.2. Physical and Chemical Characterization of Carbon Fiber–Silica Aerogel Nanocomposite

The results of basic physicochemical parameters of silica aerogel-based nanocomposites depending on the number and type of fibers, are presented in Table 2. The analysis of the results indicates that the higher number of fibers, 15 vol.%, increases the density of the composite and decreases the BET, as a result of lower volume of micropores. These tendencies were also observed in the nickel-coated carbon fibers. The presence of carbon fibers, especially nickel-coated fibers, in the structure of the aerogel positively affected the electrical conductivity. The greater number of fibers resulted in a two-fold increase of conductivity of the material, independently of the testing method. The best structural parameters were obtained for aerogel with a 10 wt.% addition of carbon fibers, and the composite was characterized by a light structure without microcracks, and with porosity amounted to 91%. For composite AG10%CF, the silica aerogel particles are in-built within the carbon fibers network, and as result, the obtained structure exhibits slightly lower density compared to pure silica aerogel. Moreover, the indirect explanation is the change in the average pore diameter and average pore volume. These are higher in the case of a composite with 10 wt.% of carbon fibers. The conducted research indicated that carbon fibers with a Ni nanolayer present in the silica aerogel structure negatively affect the structural properties of composite, but they enhanced the electrical conductivity of composite by more than two-times. This is probably associated with the blocking of interaction between the fiber surface and silica skeleton by a nanolayer of nickel deposited on the carbon fiber surface. In the case of pure carbon fibers, free hydroxide and acidic functional groups present on the surface can react with silica aerogel frame, resulting in an increase of the specific surface area and more stable structure of the composite.

The surface characterization of silica aerogel and silica aerogel nanocomposite with pure and nickel-coated carbon fibers depending on the number of fibers and using infrared spectra is presented in Figure 4. For all materials, a broad band associated with Si–O–Si bonds of the silica frame appears approx. at wavenumber 1100 cm^−1^ [35,36]. Additionally, in all cases there are bands at wavenumbers 1258 cm^−1^, 847 cm^−1^, and 758 cm^−1^ that correspond to the Si–C bond, which can be assigned to the modification of the gel in TMCS [36,37]. In case of composites with nickel-coated carbon fibers, an increased intensity of bands at frequencies ranging between 1700–1600 cm^−1^ was observed. This indicates the presence of C=O bonds in the carbonyl and lactone functional groups. Additionally, in this material, widening of the bands at frequencies range 1500–1200 cm^−1^ was observed. This can be due to the end-effect of overlapping of bands originating from the stretching and oscillatory vibrations of C–H bonds. This may indirectly indicate the lack of total reaction between the silica structure and the surface area of carbon fibers that are blocked by the nickel nanolayer and from which the increase of the silica chain begins in the case of a composite with pure carbon fibers [26,27,28].

Raman spectra of silica aerogel AG, carbon fibers CF, and nanocomposites AG10%CF, AG15%CF, AG10%CFNi, AG15%CFNi, which correspond to the FTIR analysis are seen in Figure 5. Via the Raman spectra of pure silica aerogel, and as-prepared silica aerogel composites with carbon fibers, we can see the visible characteristic bands of silica. The Raman band at 973 cm^−1^ is related to Si–OH groups, the Raman band at around 800 cm^−1^ is attributed to the SiO_2_ network, and the broad band in the range from 200 cm^−1^ to 500 cm^−1^ is associated with Si–O–Si groups [38]. In the Raman spectra of AG and AG15%CF, besides the Raman bands of silica, visible Raman bands related to TMCS can be seen. The band at 240 cm^−1^ is assigned to antisymmetric SiC_3_ deformation vibration in TMCS, and bands at 605 cm^−1^ and 617 cm^−1^ are assigned to symmetric SiC_3_ stretching vibration, while bands at 692 cm^−1^ correspond to antisymmetric SiC_3_ stretching vibration in TMCS. The other bands of TMCS at 759 cm^−1^, 846 cm^−1^, 1266 cm^−1^, 1415 cm^−1^, 1460 cm^−1^ correspond to CH_3_ bending vibration, and bands at 2855 cm^−1^, 2906 cm^−1^, and 2965 cm^−1^ are associated to CH_3_ antisymmetric and symmetric stretching vibration [39]. In the Raman spectrum of CF and CF with nickel, two characteristic bands at 1307 cm^−1^ and 1590 cm^−1^ are visible. The first one (D band) is related to the breathing modes of sp^2^ atoms in rings and corresponds to the structural defects or edges in the carbon structure. The second one (G band) is associated with the bond stretching of all pairs of sp^2^ atoms in both rings and chains [40]. D band and G band are also visible in Raman spectrum of all the study composites and they are indicated in Figure 5.

Figure 6 show the thermal analysis of pure silica aerogel and as-prepared nanocomposites with pure and nickel-coated carbon fibers. We observed that composites with 10 and 15 vol.% addition of carbon fibers are characterized by high thermal stability, indicating a 2% and 4% loss of weight at temperatures up to 400 °C, respectively. Nevertheless, both materials exhibited high thermal stability up to 700 °C, as only 6 and 10% losses of weight were noticed and in whole, the tested range exhibited higher thermal stability compared to pure silica aerogel. This confirmed the effective modification of the structure in TMCS. In case of silica aerogel with nickel-coated carbon fiber, the thermal stability in whole tested range was lower, however, the maximum weight loss equal to 20% was observed in case of 15 vol.% content of carbon fibers at 700 °C. The presence of a higher number of carbon fibers, especially nickel-coated fibers, negatively influenced the thermal stability of the nanocomposite.

Aside from the structural parameters, Table 2 includes the results of electrical conductivity of the silica aerogel and its composites with pure and nickel-coated carbon fiber at a volume of 10 and 15%. Non-conductive materials may conduct current after introducing particles or fibers with conductive properties into their structure at amounts that ensure the percolation threshold. This is a minimal amount of conductive addition that enables the flow of current. It is easier to reach the percolation threshold for fibers than for spherical particles—indeed, such is possible even in the case of a few percent addition and the effect strongly depends on the structural parameters of the fibers, but also on the properties of the matrix. For example, in case of the cement matrix, which exhibits ionic conductivity, the percolation threshold for carbon fibers ranges from 0.5 to few %, depending on the length and diameter of the fibers [41]. It is much more difficult to achieve the percolation threshold for dielectric materials, such as silica aerogels. In this case, the percolation threshold that enables the creation of a fiber net was obtained by a 10% addition of fibers.

The microstructures of the aerogel composite with 10 and 15% additions of fibers are presented in Figure 7 and Figure 8, respectively, for unmodified and nickel-coated carbon fibers. In addition, in Figure 7 and Figure 8, with EDS we can see the analysis with the particular elements distribution of the as-prepared nanocomposites. While the carbon distribution indicates fiber connections, however, parts of the fibers that had separated from each other by the aerogel structure remain inactive. The 15% addition of fibers definitely ensures good electrical conductivity—in this case, it was two times higher in comparison to the 10% fiber composite. Both the aerogel matrix with 10 and 15 vol.% of nickel-coated carbon fibers exhibited good electrical conductivity. This was two-times higher in comparison to silica aerogel with pure carbon fibers. Similar values of conductivity were reached in case of aerogel composites with the addition of the conductive polyaniline polymer [42]. Here, the highest conductivity of 0.022 mS/cm was obtained for 12 mg/mL of polyaniline, which equaled the amount of 16.5 wt.%.

### 3.3. Electromagnetic Interference Shielding

The silica aerogel-carbon fiber composite and silica aerogel–carbon fiber coated with nickel monolayer were tested in terms of the ability to attenuate electromagnetic radiation. The transmission and reflection characterization of silica aerogel-based nanocomposite with pure and nickel-coated carbon fibers are presented in Figure 9.

The maximum damping coefficient was established on the basis of curves for reflection and transmission shielding effectiveness for each bands for all tested composites. Those values are presented in Figure 10. The best damping coefficient was noted for the silica aerogel with a 15% addition of fibers in the X band and it was equal to 56 dB for a frequency of 10 GHz. In the whole band, the values for this coefficient ranged from 23, 34 dB to a maximum value of 56 dB. In the remaining bands, the damping coefficients were lower, and ranged from 20 to 35 dB. Similar dependencies were noted for silica aerogel with a 10% addition of carbon fibers; in this case, the highest damping coefficients were also obtained for band X for a frequency ranging from 8 to 13 GHz. In the whole research range, the damping results were lower than in case of a composite with a greater amount of carbon fibers. The highest damping coefficient was reached for frequency of 13 GHz. This equaled 40 dB. In the remaining ranges, its values ranged from 18 to 28 dB. The same tendencies were observed in case of a composite with nickel-coated carbon fibers. Nevertheless, in the whole research range, the damping results were lower than in case of the composite with pure carbon fibers. The best damping coefficient (42 dB) was obtained for composite AG15%CFNi and was shifted towards a lower frequency of 9 GHz, in comparison to the composite with pure carbon fibers. In contrast, in the case of AG10%CFNi, a higher value of damping coefficient was recorded for frequency 15.5 GHz. This equaled 30 dB. For all the tested materials, the electromagnetic radiation did not infiltrate the sample, it was actually completely damped, as the transmission coefficient was noted on the 0 level.

Independently from the tested nanocomposite, the tested materials mainly absorbed the electromagnetic radiation, especially within the range from 8 to 18 GHz. Furthermore, the values of absorption coefficient ranged from 80 to 90%, whereas the reflection coefficients ranged from 2 to 20 percent (see Figure 10). Carbon fibers, like other fibers characterized by high electrical conductivity, usually reflect the electromagnetic radiation. This effect was reported by Yang et al. [43,44]. In the case of the tested composites, the percolation threshold was reached for 10% addition of carbon fibers and the composite gained conductive properties. Electrical conductivity values for silica aerogels with 10 and 15% addition of carbon fibers achieved 1.213 and 2.226 mS/cm (calculated by EIS method), respectively, and were enhanced when Ni-coated carbon fibers were applied. However, the lack of distinct differences between reflection and absorption coefficients for composites differing in number of fibers indicates that the main mechanism in both materials is the absorption of electromagnetic radiation, and the aerogel matrix with high porosity in ranges from 89 to 91% is rather more responsible for the damping of electromagnetic radiation. Absorption of electromagnetic radiation can result in this case from the effect of multiple reflection. The aerogel structure is combined from fine silica particles building a chain filled with pores of diameter from 10 to 12 nm that are filled with air. An electromagnetic wave going through pores with nanometric diameter undergoes multiple reflection from the silica walls and is thus weakened. Additionally, according to the composite microstructures presented in Figure 7 and Figure 8, the fibers in additions of 10 and 15% interrupt the creation of aerogel structure and lead to the formation of free spaces of micrometric dimensions, where, additionally, the electromagnetic wave can be reflected repeatedly, both from the surface of the fibers and from the aerogel skeleton and thus be weakened. In case of composites with Ni-coated carbon fibers, despite much lower surface areas (317.4 and 295.7 m^2^/g), the damping properties of nanomaterial were satisfactory, and reached values of 30 and 42 dB. This came about because of the higher electrical conductivity of nickel-coated fibers compared to the unmodified.

Similar tendencies were observed by Thzeng. The nickel layer on the carbon fibers enhanced the shielding properties of ABS composites, in comparison with unmodified carbon fibers and Cu-covered carbon fibers. The reason for of this effect was the thin layer of metal on the surface of carbon fibers which enhanced the conductivity due to the high electrical conductivity of the used metals [33]. A similar effect was noticed by Ling [45] in porous composites based on polyetherimides PEI and graphene in the amount of 10 wt.%. During the foaming of the composite, a microporous structure with density of 0.3 g/cm^3^ was obtained, which resulted in a significant increase of the damping coefficient from 17 to 44 dB. Similar features were gained for other porous materials. The values of damping coefficients for composites AG10%CFNi, AG10%CF, AG15%CFNi, and AG15%CF with damping coefficients for chosen porous composites based on polymers and carbon materials were compared in Figure 11. For all the materials, the damping coefficients were tested at frequencies ranging from 8 to 13 GHz, which equals the X band. The specific mesoporous structure of the silica aerogel with its high specific surface results in very good damping coefficients with regard to electromagnetic radiation—which are very often better than the values obtained for foamed polymers with conductive fillers. Good results were also obtained for foamed polystyrene with carbon nanotubes [44], foamed methyl polymethacrylate both with multi-wall carbon nanotubes additionally deposited on the surface with iron oxide Fe_3_O_4_ [46], and with graphene [47]. All composites were characterized by low density and very good damping of electromagnetic radiation, mainly on the basis of the absorption mechanism.

## 4. Conclusions

Summing up the performed research, it can be assumed that:

In the electroplating process, the carbon fibers were successfully covered by an Ni-layer. This caused an increase of electrical conductivity from 11.263 mS/cm to 13.269 mS/cm—as confirmed by X-ray and EDS analyses.

-Non-modified carbon fibers incorporated within the silica aerogel matrix improved the structural parameters and thermal stability of the silica aerogel composite, while, Ni-coated carbon fibers induced a deterioration of all factors.-The percolation threshold was gained when 10 vol.% of carbon fibers was introduced to silica aerogel matrix; this amount of fiber created a conductive net in the dielectric material and electrical conductivity of nanocomposite achieved 1.213 mS/cm. Greater amount of fibers (15 vol.%) resulted in over two times higher electrical conductivity, which further was doubled by coating the carbon fiber surface with a Ni-nanolayer.-Effectiveness of shielding properties depends strongly on the composition of the silica aerogel-carbon fiber composite. Damping was mainly due to the effect of absorbing the electromagnetic radiation that achieved values close to 90% in the tested frequency range. Composite silica aerogel–carbon fiber in the amount of 15% of volume, exhibited the best characteristics of absorption. Herein, the reflection parameters damping at 56 dB in the frequency range of 8–13 GHz, was one of the best results, in comparison to literature reports of the use of lightweight porous composites with electromagnetic shielding properties.

## Figures and Tables

**Figure 1 materials-13-00400-f001:**
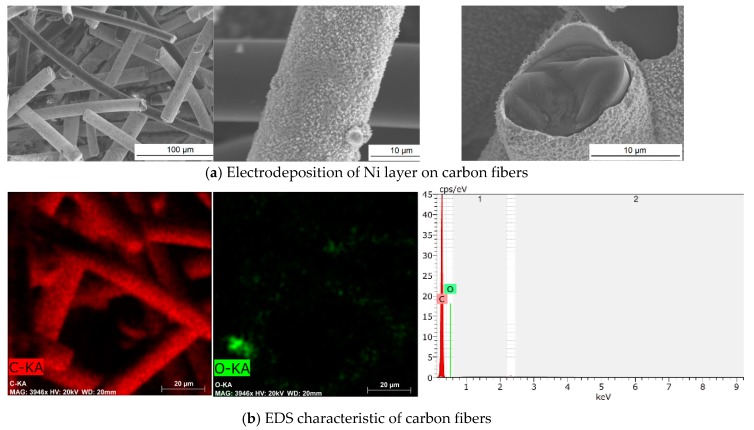
SEM images of Ni-coated carbon fibers (**a**) and EDS analysis of unmodified carbon fibers (**b**) and of carbon fibers coated with Ni nanolayer (**c**).

**Figure 2 materials-13-00400-f002:**
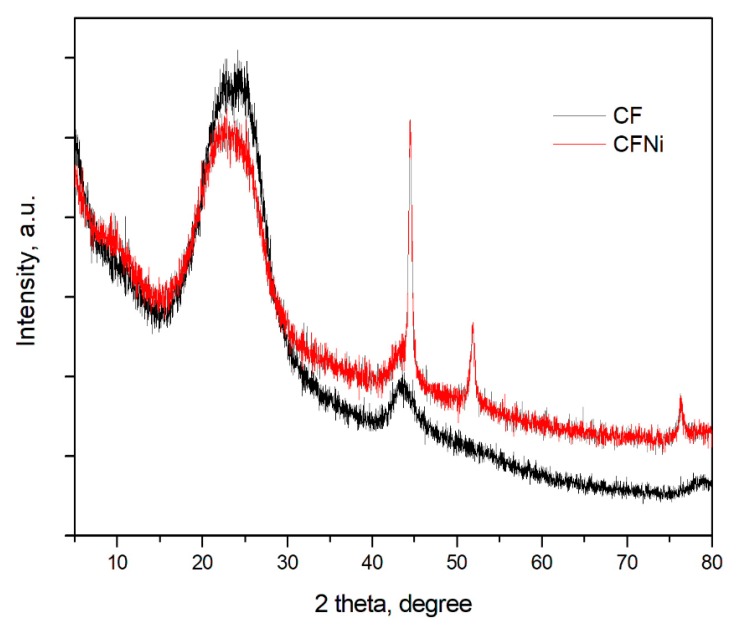
X-ray diffraction patterns for unmodified and Ni-coated carbon fibers.

**Figure 3 materials-13-00400-f003:**
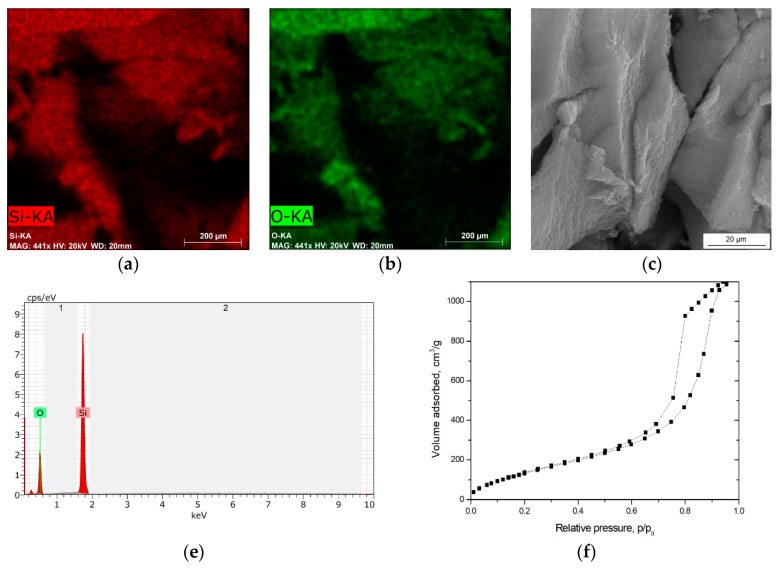
(**a**–**e**) SEM images and EDS analyses of silica aerogel; (**f**) Nitrogen adsorption-desorption isotherms at 77 K.

**Figure 4 materials-13-00400-f004:**
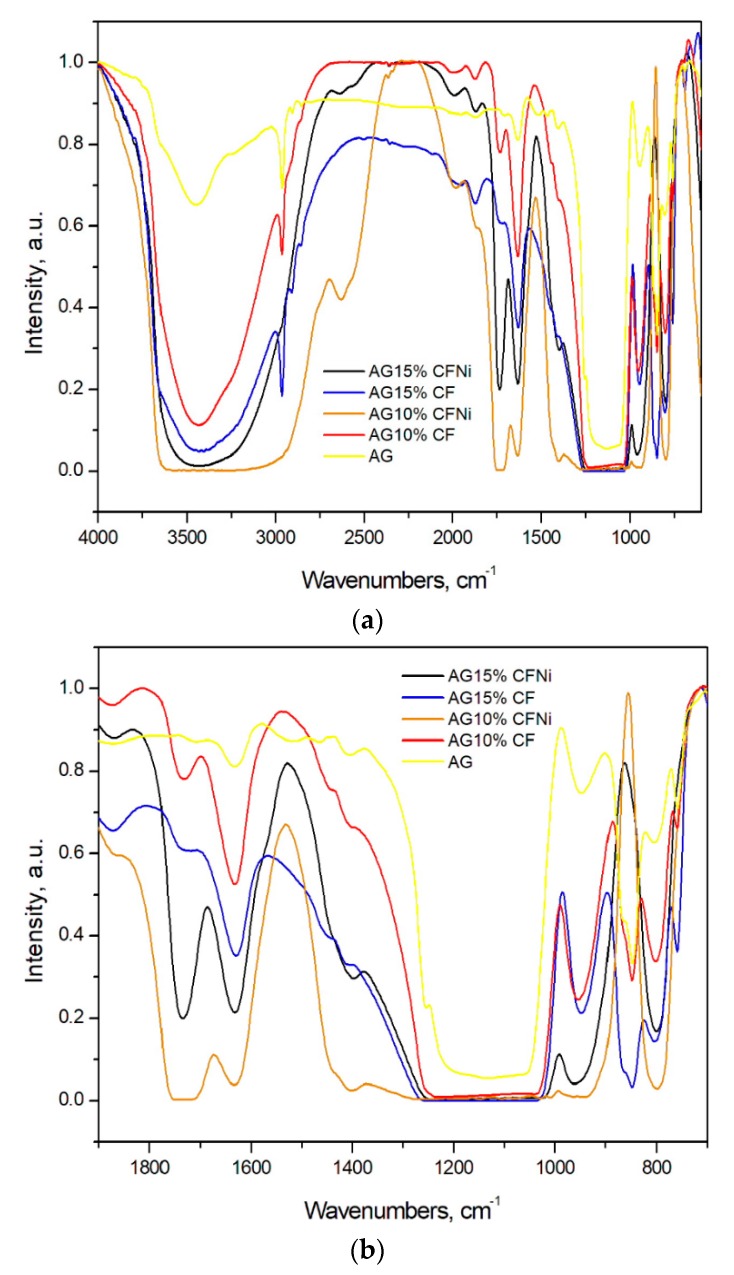
(**a**,**b**)—FTIR characterization of silica aerogel and carbon fiber–silica aerogel nanocomposites.

**Figure 5 materials-13-00400-f005:**
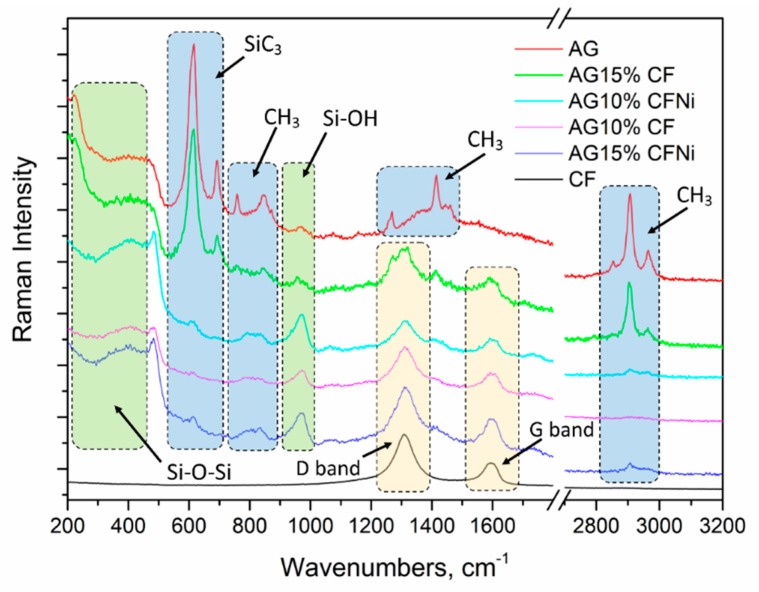
Raman characterization of carbon fibers, silica aerogel and carbon fiber–silica aerogel nanocomposites.

**Figure 6 materials-13-00400-f006:**
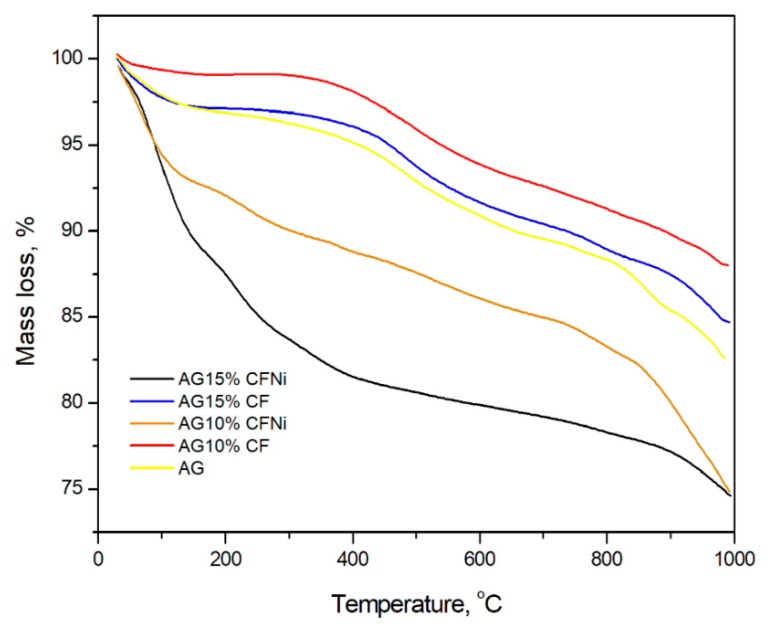
Thermogravimetric (TGA) characterization of silica aerogel and carbon fiber–silica aerogel nanocomposites.

**Figure 7 materials-13-00400-f007:**
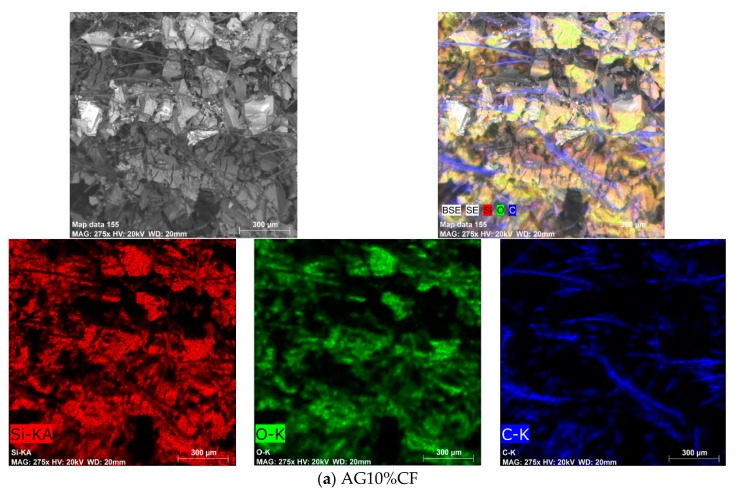
SEM images and EDS analysis of silica aerogel–carbon fibers microstructure for as-prepared nanocomposite AG10%CF (**a**) and AG15%CF (**b**).

**Figure 8 materials-13-00400-f008:**
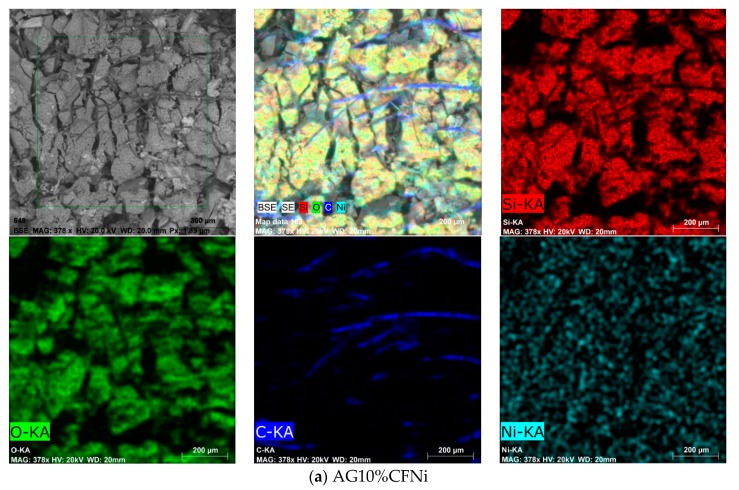
SEM images and EDS analysis of silica aerogel–carbon fiber microstructure for as-prepared nanocomposite AG10%CFNi (**a**) and AG15%CFNi (**b**).

**Figure 9 materials-13-00400-f009:**
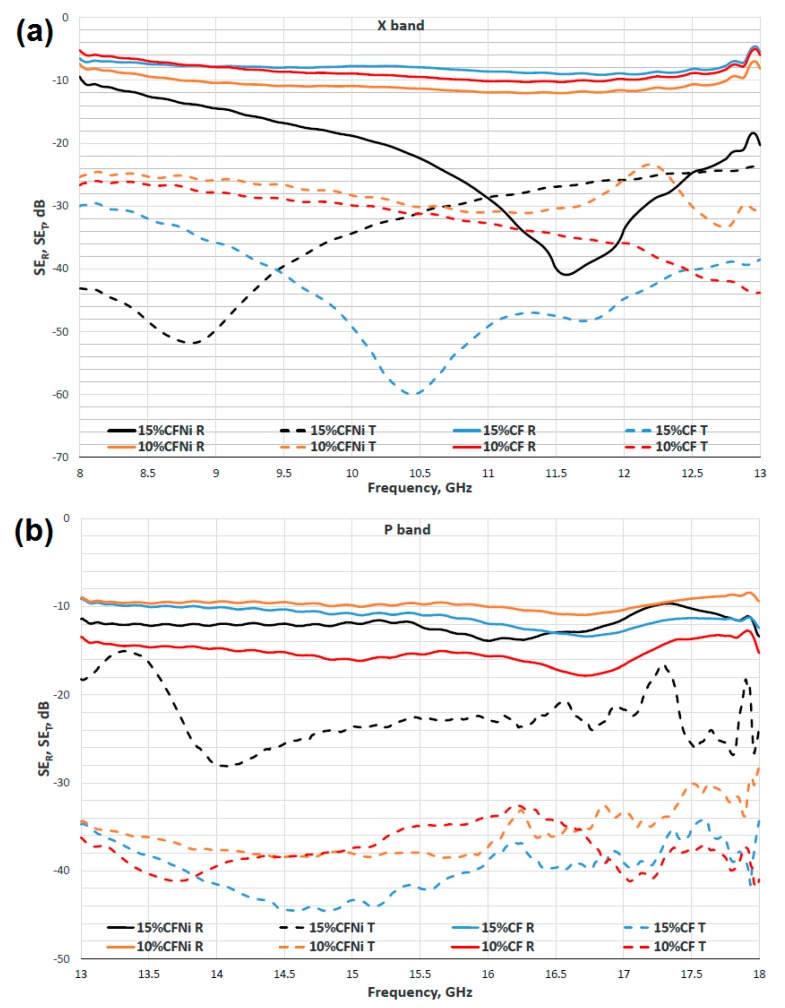
Characterization of shielding effectiveness for reflection R and transmission T mechanism in X band (**a**) and P band regions (**b**) of carbon fiber–silica aerogel nanocomposites.

**Figure 10 materials-13-00400-f010:**
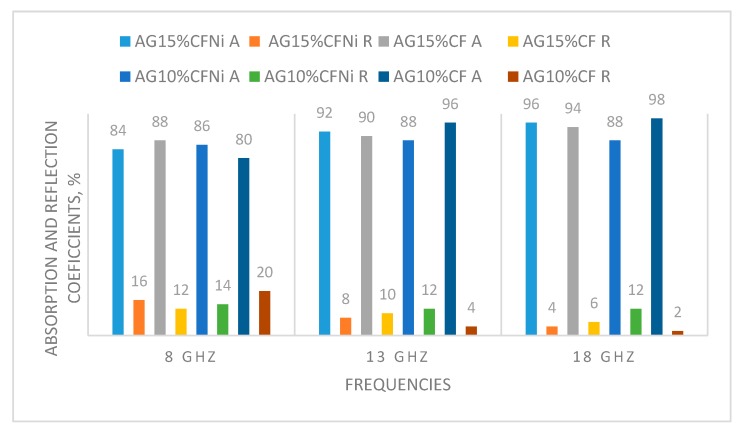
Comparison of absorption and reflection coefficients for selected frequencies for received composites in relation to the type and amount of carbon fibers.

**Figure 11 materials-13-00400-f011:**
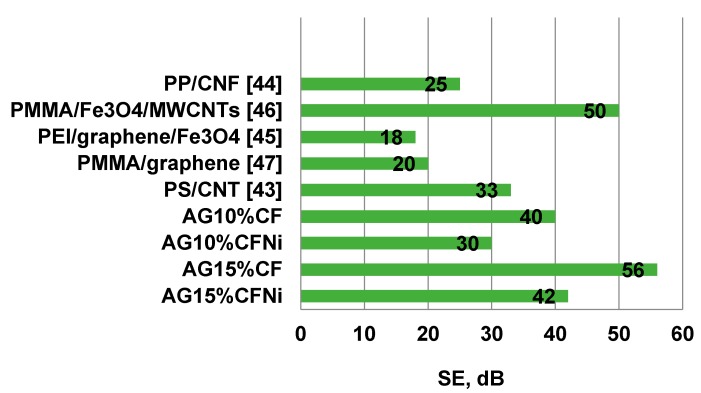
Comparison of damping coefficients for composites received in present paper and other selected lightweight composites reinforced with carbon materials.

**Table 1 materials-13-00400-t001:** Physical and structural characterization of pure silica aerogel.

Material	Form	Density, g/cm^3^	Porosity, %	Specific Surface Area, m^2^/g	Average Pore Diameter, nm	Average Micropore Volume, cm^3^/g
Pure AG	granulate	0.201	90.9	496.5	10.2	1.27

**Table 2 materials-13-00400-t002:** Physical and structural characterization of carbon fiber–silica aerogel nanocomposites.

Parameter	AG15%CFNi	AG15%CF	AG10%CFNi	AG10%CF
Form	monolith	monolith	monolith	monolith
Density, g/cm^3^	0.270	0.225	0.248	0.199
Porosity, %	88.0	90.0	89.0	91.0
Specific surface area, m^2^/g	295.7	467.0	317.4	474.6
Average pore diameter, nm	8.4	12.7	13.6	14.5
Average micropore volume, cm^3^/g	0.561	1.486	0.996	1.724
Conductivity, mS/cm (LSV method)	4.602	2.019	3.726	1.065
Conductivity, mS/cm (EIS method)	4.820	2.226	3.722	1.213

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
