# Peer review of "Carbon Fiber and Nickel Coated Carbon Fiber–Silica Aerogel Nanocomposite as Low-Frequency Microwave Absorbing Materials"

_materials, 2020, doi:10.3390/ma13020400_

Round 1

Reviewer 1 Report

In this manuscript, authors have reported their experimental data in an effort to prepare the water glass based silica aerogels integrated with carbon fibers and/or nickel coated carbon fibers as potential low-frequency microwave absorbing materials. They have demonstrated that the prepared materials exhibit large electromagnetic radiation absorption capacity. However, some key information (such as porosity) is missing or misleading, and therefore, the manuscript cannot be accepted at its current statue. If authors are willing to make revision, the following are detailed comments and suggestions.

(1) Authors are required to provide evidence to confirm the formation of water glass based silica aerogel before introducing carbon fibers and/or nickel on carbon fibers (primitive silica aerogel). Authors are required to describe how they measured the bulk density and should calculate the porosity of the materials based on the density obtained. These values can be included in Table 1 for comparison (total 5 samples by adding primitive silica aerogel data).

(2) Results and discussion section, the subtitle “3.1. Chemical Characterization of Carbon Fibres and Nickel-Coated Carbon Fibres” does not make sense. The only paragraph under this title describes the surface morphologies. In addition, no information is given for the figure captions (d, e) in Fig. 1. It is strongly suggested to include the SEM images of primitive silica aerogel in Fig. 1 for comparison. Furthermore, to confirm the formation of uniformed Ni coating layer on carbon fiber surface, it is preferable to provide the cross-sectional SEM image.

(3) Figure 2 does not show the detailed features in the spectrum range that authors mentioned. Please provide the enlarged FTIR spectra at < 2000 cm-1 for more meaningful description related to the paragraph (Line 198 to 208). The FTIR spectrum of primitive silica aerogel is also needed and should be included in Fig. 2 for comparison.

(4) Similarly, the SEM image in Fig. 4 should be compared with that of primitive silica aerogel at the same scale.

(5) Some key information is missing or misleading. It is indeed understandable the apparently enhanced conductivity achieved by adding carbon fibers and/or nickel coated fibers into almost insulated silica aerogel (sacrifice of porosity). Authors should explain whether carbon fibers or metallic nickel coating would enhance the conductivity/electromagnetic radiation absorption capacity at the same way. The information about physical and chemical properties of primitive silica aerogel is necessary. Please supply this information throughout the manuscript (from introduction to conclusion).

(6) Conclusions are too long. Please revise it.

Author Response

Dear Reviewer,Thank You for Your insightful review of our work, which contributed to a better understanding of the scientific problems related to the subject of the publication and will help with the elimination of potential errors in the future.We would also like to express our gratitude for the revision of our manuscript and the opportunity to re-submit it, incorporating all of the Referees’ suggestions. Our comments and changes are noted below, and were marked in yellow in the manuscript. 

In this manuscript, authors have reported their experimental data in an effort to prepare the water glass based silica aerogels integrated with carbon fibers and/or nickel coated carbon fibers as potential low-frequency microwave absorbing materials. They have demonstrated that the prepared materials exhibit large electromagnetic radiation absorption capacity. However, some key information (such as porosity) is missing or misleading, and therefore, the manuscript cannot be accepted at its current statue. If authors are willing to make revision, the following are detailed comments and suggestions.

(1) Authors are required to provide evidence to confirm the formation of water glass based silica aerogel before introducing carbon fibers and/or nickel on carbon fibers (primitive silica aerogel). Authors are required to describe how they measured the bulk density and should calculate the porosity of the materials based on the density obtained. These values can be included in Table 1 for comparison (total 5 samples by adding primitive silica aerogel data).

Answer: We thank the Reviewer for this valuable comment. The changes have been made in the manuscript and are marked in yellow. Additionally we completed the structural parameters of pure silica aerogel and silica-aerogel nanocomposite like average micropore volume and pore diameter. 

(2) Results and discussion section, the subtitle “3.1. Chemical Characterization of Carbon Fibres and Nickel-Coated Carbon Fibres” does not make sense. The only paragraph under this title describes the surface morphologies. In addition, no information is given for the figure captions (d, e) in Fig. 1. It is strongly suggested to include the SEM images of primitive silica aerogel in Fig. 1 for comparison. Furthermore, to confirm the formation of uniformed Ni coating layer on carbon fiber surface, it is preferable to provide the cross-sectional SEM image.

Answer: We thank the Reviewer for this valuable comment. We revised the characterization of carbon fibers and nickel-coated carbon fibers and all changes have been made in the manuscript and are marked in yellow.  

(3-4) Figure 2 does not show the detailed features in the spectrum range that authors mentioned. Please provide the enlarged FTIR spectra at < 2000 cm-1 for more meaningful description related to the paragraph (Line 198 to 208). The FTIR spectrum of primitive silica aerogel is also needed and should be included in Fig. 2 for comparison.

Similarly, the SEM image in Fig. 4 should be compared with that of primitive silica aerogel at the same scale.

Answer: We thank the Reviewer for this valuable comment. The changes have been made in the manuscript and are marked in yellow. Additionally we completed the TG data presented in Fig. 6 with thermal stability of pure silica aerogel. 

(5) Some key information is missing or misleading. It is indeed understandable the apparently enhanced conductivity achieved by adding carbon fibers and/or nickel coated fibers into almost insulated silica aerogel (sacrifice of porosity). Authors should explain whether carbon fibers or metallic nickel coating would enhance the conductivity/electromagnetic radiation absorption capacity at the same way. The information about physical and chemical properties of primitive silica aerogel is necessary. Please supply this information throughout the manuscript (from introduction to conclusion).

Answer: We thank the Reviewer for this valuable comment. The changes and explanation of carbon fibers functioning in silica aerogel matrix have been made in the manuscript and are marked in yellow.

(6) Conclusions are too long. Please revise it.

Answer: We thank the Reviewer for this valuable comment. We shortened the conclusions and the changes have been made in the manuscript and are marked in yellow. 

We look forward to hearing from you.

Yours faithfully,

Dr. Agnieszka Ślosarczyk,

Poznan University of Technology

Reviewer 2 Report

This paper describes a Carbon fibres and nickel coated carbon fibres–silica 
aerogel nanocomposite as low-frequency microwave absorbing material .The results are interesting. I think it can be consider to publish if the following issues are solved.

the basic data like xrd should be provided and the Ni peak and the carbon fibers should be proved by it. you can refere the paper like: Journal of Catalysis 379 (2019) 154–163; Applied Catalysis B: Environmental 240 (2019) 92–101; J. Mater. Chem. A, 2018,6, 20304-20312. the optimum loading percentage should be clarified and the stability should be added. like : Chemical Engineering Journal 369 (2019) 353–362; Journal of Catalysis 361 (2018) 238–247; Journal of Catalysis 355 (2017) 1–10; Carbon 122 (2017) 287-297.  it will be better if the raman and xps data is provided. the english level can be improved.

Author Response

Dear Reviewer,Thank You for Your insightful review of our work, which contributed to a better understanding of the scientific problems related to the subject of the publication and will help with the elimination of potential errors in the future.We would also like to express our gratitude for the revision of our manuscript and the opportunity to re-submit it, incorporating all of the Referees’ suggestions. Our comments and changes are noted below, and were marked in yellow in the manuscript. 

This paper describes a Carbon fibres and nickel coated carbon fibres–silica 
aerogel nanocomposite as low-frequency microwave absorbing material. The results are interesting. I think it can be consider to publish if the following issues are solved.

The basic data like xrd should be provided and the Ni peak and the carbon fibers should be proved by it. you can refere the paper like: Journal of Catalysis 379 (2019) 154–163; Applied Catalysis B: Environmental 240 (2019) 92–101; J. Mater. Chem. A, 2018,6, 20304-20312. the optimum loading percentage should be clarified and the stability should be added. like : Chemical Engineering Journal 369 (2019) 353–362; Journal of Catalysis 361 (2018) 238–247; Journal of Catalysis 355 (2017) 1–10; Carbon 122 (2017) 287-297. 

Answer: We thank the Reviewer for this valuable comments. We provided the XRD data for carbon fibers and nickel-coated carbon fibers. The changes have been made in the manuscript and are marked in yellow. In addition we extended the fibers and silica aerogel-based nanocomposites characterization with the EDS analysis.

It will be better if the raman and xps data is provided.

Answer: We thank the Reviewer for this valuable comments. We provided the Raman data for carbon fibers and silica aerogel composites. The description of Raman spectroscopy results were added in the text. The changes have been made in the manuscript and are marked in yellow. XPS analysis we will take into account in the future analysis.

The english level can be improved.

Answer: The whole manuscript has been carefully checked with regard to editorial and language issues.

We look forward to hearing from you.

Yours faithfully,

Dr. Agnieszka Ślosarczyk,

Poznan University of Technology

Reviewer 3 Report

Interesting work and a well written publication. A number of suggestions:

Figure 10: This Figure is very difficult to interpret. The y-axis is not labelled, however you do describe the plot in the caption, that said, they appear to be presented as percentages, so if this is the case, the axis should be labelled (absorption and reflection coefficients are not normally percentages per your description associated with Equation 2 (R+A+T=1)). I would also suggest changing the color scheme or directly labelling the bars as the colors are very difficult to distinguish.

Lines 419- 422: Is this referring to another study (Cu-covered carbon fibres in an ABS polymer? If so it needs to be referenced.

Figure 11: Include references for the reported results in the Figure. If you have done this in the paragraph before the Figure it is not clear and would be better incorporated into the Figure itself for clarity. To enhance this Figure (and your results further) it would be good to include the densities of the respective materials in Figure 11 if available.

Author Response

Dear Reviewer,

Thank you very much for your valuable comments and the possibility to resubmit the article.

The suggestions about figures and literature have been introduced in the text and you could see the revision track.

We look forward to hearing from you.

Yours faithfully,

Dr. Agnieszka Ślosarczyk,

Poznan University of Technology

Round 2

Reviewer 1 Report

Authors have replied the most comments and suggestions from the reviewer. However, after they added the Pure AG data in the New Table 1, the problem became obvious. The bulk density of AG10%CF (0.199 g/cm3 in the revised Table 2) was even samller than that of Pure AG (0.201 g/cm3, in New Table 1), which led to a slight larger porosity (91.0% compared with 90.0% of Pure AG). This is not generally expected. Authors should double check their data or measuring procedures from Pure AG (It is very important as the data are used for comparison!) and AG10%CF. They should also provide a detailed explannation with experimental evidence if this is a case. In addition, authors have shown the EDS analysis results in both spectrum and mapping image forms (e.g. in Fig. 1 and Fig. 3). They should include these information in the text and the caption. Finally, English needs to be polished and the apparent typo or grammar errors must be corrected. For example, Line 149-150, the sentence "In addintion the a PANalytical X-ray diffractometer was used in the X-ray analysis with......" may be revised to "In addition, a PANalytical X-ray diffractometer was used to analyze the crystal structres of  carbon fibers and metal coatings with ......".

Author Response

Dear Reviewer,

Thank you very much for your valuable comments and the possibility to resubmit the article.

Authors have replied the most comments and suggestions from the reviewer. However, after they added the Pure AG data in the New Table 1, the problem became obvious. The bulk density of AG10%CF (0.199 g/cm3 in the revised Table 2) was even samller than that of Pure AG (0.201 g/cm3, in New Table 1), which led to a slight larger porosity (91.0% compared with 90.0% of Pure AG). This is not generally expected. Authors should double check their data or measuring procedures from Pure AG (It is very important as the data are used for comparison!) and AG10%CF. They should also provide a detailed explannation with experimental evidence if this is a case. In addition, authors have shown the EDS analysis results in both spectrum and mapping image forms (e.g. in Fig. 1 and Fig. 3). They should include these information in the text and the caption. Finally, English needs to be polished and the apparent typo or grammar errors must be corrected. For example, Line 149-150, the sentence "In addintion the a PANalytical X-ray diffractometer was used in the X-ray analysis with......" may be revised to "In addition, a PANalytical X-ray diffractometer was used to analyze the crystal structres of  carbon fibers and metal coatings with ......".

The presented results of silica aerogel and silica aerogel – carbon fibers nanocomposite are correct. The reason of the difference is the particle size distribution in pure aerogel which is in granulate form, and in the same volume the higher number of particles are contain, compared to composite with carbon fibers. Generally, in this case, pure silica aerogel during ambient pressure drying decomposes to the xerogel form. In case of nanocomposite the silica aerogel particles are in-built in carbon fibers network, and as result the obtained structure with 10 wt.% exhibit slightly lower density. Moreover, the indirect explanation is the change in the average pore diameter and average pore volume which are higher in case on composite with 10 wt.% of carbon fibers than for pure silica aerogel. The application of modified carbon fibers lead to the chemical reaction between carbon fiber surface and silica aerogel and creation more stable structure with better structural parameters than in case of pure silica aerogel. Besides based on SEM observation nanocomposite with carbon fibers has some micropores connected with the way of synthesis, which lead to diminishing the composite density.

The above suggestions have been introduced in the text and were marked in yellow. The text was checked under the spelling and grammar errors by native speaker.

We look forward to hearing from you.

Yours faithfully,

Dr. Agnieszka Ślosarczyk,

Poznan University of Technology

Reviewer 2 Report

It can be accepted now.

Author Response

Dear Reviewer,

Thank you very much for your valuable comments and the possibility to submit the article.

Yours faithfully,

Dr. Agnieszka Ślosarczyk,

Poznan University of Technology

Reviewer 3 Report

I am happy with the improvements made. Great work!